# Influence of Teacher and Family Support on University Student Motivation and Engagement

**DOI:** 10.3390/ijerph18052606

**Published:** 2021-03-05

**Authors:** Adela Descals-Tomás, Esperanza Rocabert-Beut, Laura Abellán-Roselló, Amparo Gómez-Artiga, Fernando Doménech-Betoret

**Affiliations:** 1Developmental and Educational Psychology, Universitat de València, 46010 Valencia, Spain; Esperanza.Rocabert@uv.es (E.R.-B.); Amparo.Gomez@uv.es (A.G.-A.); 2Developmental and Educational Psychology, Universitat Jaume I, 12071 Castellón, Spain; labellan@uji.es

**Keywords:** teaching support, family support, expectancy beliefs, value beliefs, achievement goals, student engagement

## Abstract

Although many studies endorse the notion that the way students perceive support influences their engagement, very few have explored the possible mediator role of intention to learn between these variables. The present work provides new evidence to the existing literature because it analyses the work of intention to learn (measured with expectancy–value beliefs and achievement goals) as a mediating motivational variable in the relation between university students’ external support (teacher and family) and their engagement. The Educational Situation Quality Model (MOCSE, its acronym in Spanish) has employed as a theoretical framework to perform this analysis. A sample of 267 Spanish university students completed the questionnaires employed to measure the considered variables at three times. They answered teacher and family support scales when the course began (time 1), intention to learn scales halfway through the course (time 2), and engagement scales when the course ended (time 3). The obtained structural equation models showed a positive and significant effect for teacher and family support on the considered motivational variables (expectancy–value beliefs and achievement goals) and these, in turn, on student behavioral engagement. These results allow us to point out a series of recommendations for university teachers to improve their students’ involvement in their learning process.

## 1. Introduction

When a student engages more than others in learning a given subject, what does this depend on? What can a teacher and family do to help students engage in and enjoy learning a given subject? This study attempts to shed light on the answers to both these questions. The main objective of this research work is to examine the relations among external support (teacher and family), relevant motivational variables (expectancy–value beliefs and achievement goals), and the degree of involvement (engagement) adopted by students. Understanding why a student decides to participate/implicate or not in the teaching–learning (T–L) process is crucial to take educational actions that help to improve their learning outcomes.

Although a large body of literature on this field exists, for teacher [1,2,3,4] and family support [5,6], very little research has analyzed the mediator role played by socio–cognitive theories of motivation (e.g., expectancy–value and achievement goal theories) in the relation between teacher–family support and student engagement.

The present study addresses this analysis from a new perspective by using the Educational Situation Quality Model (MOCSE, its acronym in Spanish) of Doménech-Betoret, [7,8,9,10] as a reference framework.

The MOCSE is an instructional model that explains how an educational setting works with an integral approach. Although the MOCSE simultaneously considers the T–L process, this work centers on students’ learning process: the academic demands and support perceived by students to meet these demands (Stage I) trigger the intention to learn (Stage II), which, in turn, affects students’ behavior and emotions during the learning process (Stage III) which, finally, impact learning outcomes, such as student satisfaction and their achievements (Stage IV). Figure 1 represents the sequence of students’ actions for one course according to the MOCSE postulates.

MOCSE is not simply a theoretical model that aims to guide research in the educational context, like most of the instructional models found in the literature, but also provides a methodological procedure for teachers to apply it in the classroom. So more than a conceptual framework, it is also a useful tool that provides the keys and systematic guidelines (evaluation, reflection, intervention) so that teachers are able to diagnose their students’ motivational deficiencies and to identify their possible causes in terms of demands and supports/resources. It is valuable information for designing improvement actions to correct detected motivational deficiencies and to improve academic outcomes. Improvement actions can be implemented either during the course underway or the next course (for more details, see [9]). Another specific advantage of the model for studying school engagement is that the MOCSE, based on the theory of action control [11], distinguishes between predecisional motivation (desire, intention) and postdecisional motivation (executive or volitional motivation). This distinction allows preventive measures centering on students’ anticipatory cognitive motivators (e.g., expectations) to be adopted (for details, see [9]).

When studying how the education context works, the MOCSE integrates contributions from several relevant psycho–educational theories. The present work considers the *Job Demands–Resources Model* (JD-R) [12], the *Expectancy-Value Theory* [13], and the *Achievement Goal Theory* [14,15,16,17]. By including the constructs derived from these theories, it is possible to better understand why some university students decide to engage more than others during the T–L process followed for a given subject. In line with this, and based on the MOCSE framework, the main purpose of this study is to bridge the job demands–resources model and two dominant theories of achievement motivation in education, such as expectancy–value and achievement goal theories, by examining (a) how students’ perceptions of support/resources (from family and teacher) and students’ motivational beliefs derived from the expectancy–value theory relate to student engagement; (b) how students’ perceptions of support/resources (from family and teacher) and students’ motivational beliefs derived from the achievement goal theory relate to student engagement.

The findings of this research line can be very useful for teachers to improve students’ participation/implication (engagement) and learning outcomes. 

Figure 1 presents an overview of the general framework of the model centered on students. This general framework was adapted to study student engagement based on the MOCSE postulates and by considering previous proposals in this field [18,19] as references. Please see Figure 2.

Below, the constructs considered in this study are briefly explained. For a deeper understanding of the model, please see [9].

### 1.1. Stage I: Students’ Perception of Support Resources: Teacher and Family Support

If the Job Demands–Resources Model (JD–R) [12] is applied to the academic context, it should be assumed that how university students perceive external support (teacher and family) to meet the expected demands will impact their level of motivation.

With respect to the resources/support provided by teachers, despite finding barely any coherence among authors with the employed terminology, this can be distinguished between instructional/instrumental support (related to facilitate students’ competence) and affective/emotional support (related to favor the relationship with students). Different authors [20,21] consider instructional/affective support to form part of one single dimension, which they call “teacher participation”.

The instructional support offered by teachers intends to help students to master contents so they can meet learning demands. For the present study, the following teachers’ instructional support for their students were selected: formative evaluation (offering feedback) and competence support (encouraging students’ self-competence). 

Affective teacher support intends to meet students’ requirements and psychological desires in the classroom by contributing to trigger positive emotions and to create a healthy classroom climate. The affective support types selected for this study were motivational support (teacher’s effort to motivate students) and relational support (the teacher establishing an open relationship with students).

The resources/support provided by the family are the closest source to social support for students in both infancy and adolescence. The literature [6,7,8,9,10,11,12,13,14,15,16,17,18,19,20,21,22] suggests reducing parents’ influence during Secondary Education because students tend to seek more support from classmates. Nevertheless, as Spera [22] points out, parents’ support is necessary for students’ academic success in all subjects, periods, and demographic groups. 

Empirical studies focusing on families have proven the significant role of parents acting as contributors to school engagement and student performance at school [23]. Family members who provide academic (e.g., assisting with homework) or motivational (e.g., recognizing effort and enhancing progress) support help to improve students’ academic performance [5]. Although most studies on family support have been carried out with Primary [24] and Secondary Education students [5], the present work states that family support may also influence university student motivation and, in turn, students’ involvement (engagement) during the T–L process.

Studies in this field indicate a wide variability as to how they conceptualize and measure parents’ behavior [25]. However, it is possible to distinguish between the academic support (e.g., help with a learning task, collaborating in information seeking, ensuring that students have time and space to study, etc.) and affective support (e.g., acknowledging effort, giving praise, providing encouragement to meet educational targets, talking about how classes are going, etc.) that parents and/or family relations offer students. Some proposals [20,26] add another family support dimension that is associated with academic motivation: structure, which refers to the rules, guidelines, and expectancies that parents set for their offspring; Chen, et al. [27] also contemplate rules or limits in the family setting and relate them positively to student participation in the academic context. In the present work, academic support, affective support, and rules at home were selected to represent the family support offered to university students.

### 1.2. Stage II: Intention to Learn Measured by Expectancy–Value Beliefs and Achievement Goals

(a) Expectancy–value Theory: variables considered

Based on the Expectancy–value Theory [13], in the present work we considered students’ beliefs in how competent they perceived themselves to be to learn a given subject and the value that this subject has for them. We added two less studied constructs in the context of the afore-cited theory: Process expectancy (enjoyment) and Control expectancy (control over results).

Success expectancy (Will I be successful in this subject?). According to Eccles & Wigfield [13] (p. 119), success expectancy refers to “individuals’ beliefs about how well they will do in upcoming tasks” and, therefore, look toward the future compared with simple self-perceptions of competence. According to Bandura [28,29], success expectancy is generally broken down into self-efficacy and outcome expectancy; the former is defined as an individual’s belief in his/her own capacity to perform a given task (a subject in our case), while the latter is defined as beliefs in investing efforts that lead to the desired outcome.

Process expectancy (How will I feel when I study this subject?). Expectancy–value theorists [30,31] consider this affective component of expectancy crucial given the importance of students’ affective state for their participation while they learn. It refers to the affective feelings or reactions that students expect to feel throughout the course, which derive from teacher–student relationships and content–students-classmates.

Control expectancy (What do the outcomes obtained in this subject depend on?). Control expectancy refers to beliefs in being in control of the academic outcomes that students expect to obtain in the future and have been taken as relevant variables for student engagement and their academic performance [32,33]. Students may consider that their outcomes depend on either the variables they can control (the way they study, effort, time invested, etc.) or those they have no control over (luck, type of questions the teacher includes in exams, selected themes, etc.). During the T–L process, it is fundamental that the teacher leads expectancy toward students controlling the learning process and outcomes so they believe that fulfilling a certain goal in the future will depend on their own actions.

Task value (What value does the subject have for me?). Students’ beliefs in it being worthwhile to follow a task or theme are important for them to understand their behavior and learning outcomes [29]. The present work employs four value components that derive from the modern Expectancy–value Theory [13,34] so that students assess the value that a subject has for them: usefulness, importance, interest, and cost components.

(b) Achievement Goal Theory: variables considered

The researchers of this theory tend to consider three achievement goals that students can adopt [35,36]: Mastery goals (they centre on acquiring competence for a task or theme, e.g., “I intend to learn as much as possible”), Performance goals (worried about demonstrating their competence to others; e.g., “I try to perform better than other students”); Performance-avoidance goals (wishing to avoid social judgments made and humiliation shown by others, such as teachers or classmates, e.g., “I make an effort to avoid obtaining worse results than others”).

Most of the research conducted in this field has concentrated on Mastery and Performance goals [37]. However, as King & McInerney [20] point out, several researchers defend the idea that other goal types must be analyzed because students can adopt many goals while learning. Scientific empirical evidence can be found on this matter (see [8]), which supports the need to consider *work-avoidance goals* (students who focus on making the minimum effort to learn, e.g., “I choose easy options in class so I don’t have to work hard”) given their negative impact on student engagement and achievement. By considering the role they may play in engagement, our study includes the *Self-worth goals* [38] that students adopt when they feel proud of their own performance.

### 1.3. Stage III: Student Engagement

Engagement is one of the most highlighted themes in current education research given its importance for academic outcomes and academic success. Nevertheless, authors have not reached a consensus about how to define this construct [39] and, consequently, as to how it must be measured. Generally speaking, in the academic context, engagement takes place when students participate in the learning of academic tasks and is characterized by making continuous efforts and showing determination and perseverance in learning [29]. Avoidance strategies are considered a negative indicator of student engagement because students adopt them when they feel defeated about, drop out of or disconnect from a learning task related to a specific theme.

In recent years, student engagement has been considered a multidimensional construct that deals with a variety of dimensions related to participation at school or engagement with learning [40,41,42]. Most research works analyze student behavioral engagement, but cognitive engagement [43], social engagement, and even emotional engagement [44], are also contemplated.

This study centered on Higher Education by examining the behavioral dimension of student engagement because it is the most observable one and the easiest one to measure. Behavioral engagement refers to how students behave. It includes the actions performed by students and the efforts they invest [40,45], such as asking questions, actively participating in class, paying attention and making notes, participating in academic and/or extracurricular activities, doing voluntary work, etc. This work evaluates this behavioral engagement dimension through behaviors related to the attention, persistence, and dedication that students show throughout the T–L process.

### 1.4. Relations among Variables

Most studies that relate the variables herein considered have been conducted with Secondary Education students and, therefore, very little is known about the relation of these variables with university students. Many research works have centered on analyzing how they directly affect the support perceived by students in their engagement conducts. The findings of these works [46,47,48,49,50,51,52,53] indicate that student engagement is associated with their perceptions of teacher and/or family support and consistently evidence a positive relation between them, in such a way that the greater the perceived support from teachers and family, the greater the student academic engagement.

The present work add to the existing literature with new pieces of evidence in this field, because the direct relations between social support and student engagement are not analyzed as usual. Nowadays, the need to investigate the possible mediator or modulator role that other (personal or academic) variables play between perceived support and student engagement [54] is acknowledged. Accordingly, the mediator role played by expectancy–value beliefs and achievement goals (variables relating to students’ intention to learn) between perceived support and student behavioral engagement is specifically examined herein. The present work now goes on to provide evidence for the relation among these variables.

#### 1.4.1. Support Resources and Expectancy–Value Beliefs and Student Engagement

As far as we know, very little is known about how students’ Expectancy–value beliefs are related to the support perceived by students and engagement. Findings about the influence of social support on Expectancy–value beliefs are limited. Simpkins et al. [25] considered that parental support predicts students’ motivational beliefs; taking a positive attitude toward learning, collaborating in learning tasks, and worrying about academic tasks are all parents’ conducts that predict greater self-concept and the value of science subjects for the adolescents recruited for their study. The results of other research works also point out that parental support and participation positively influence how their offspring value the subjects they learn at school [55].

Regarding the relation between teacher support and students’ motivational beliefs, different research works have related positive teacher support to adaptive motivational results, such as evaluating academic tasks more highly, showing more interest, and greater self-efficacy (see [20]). Dietrich et al. [56] point out that many empirical findings indicate a clear positive association between teacher support and the intrinsic value that students give a task. Regarding teacher support dimensions, Ruzek et al. [57] stress that research offers consistent findings that link emotional teacher support with motivation and student self-reported engagement.

Considerable empirical support for the relation between Expectancy–value beliefs and student engagement can be found for Secondary Education [58,59,60] and University students [61,62], which confirms that beliefs are significant predictors of engagement. The results of some studies show that Expectancy–value beliefs strongly influence performance, whereas the subject value of a task considerably impacts choice, effort, and persistence [63,64].

#### 1.4.2. Support Resources and Achievement Goals and Student Engagement

Very few research works have simultaneously contemplated the influence of perceived support in students’ achievement goals and engagement. A study by Wentzel et al. [65] analyzed the three variables at the same time and found that perceived parental support was related to greater academic motivation and to positive goal orientation, which affected Secondary Education student cognitive engagement. Regarding the relation between teacher support and achievement goals, Wentzel et al. [65] stresses that several works have related adolescents’ perceptions of positive emotional teacher support with positive motivating outcomes, including the search for goals to learn. These authors assert that positive learning outcomes (e.g., achieve outcomes) also increase when other teacher support dimensions such as instructional support are considered. For University education, Senko & Dawson [66] observed a high correlation between received help and achievement goals for university students.

Finally, recent studies have found empirical evidence when relating achievement goals to engagement with Secondary Education [67,68,69] and university [70,71] students.

### 1.5. Objectives and Hypotheses

Based on the MOCSE postulates [7,8,9], the main objective of this research was to examine the relations among undergraduate students’ support (teacher–family), relevant motivational variables (expectancy–value beliefs and achievement goals), and behavioral engagement. To do so, some research questions were addressed and examined: (a) “How do undergraduate students’ perceptions of the support they are provided with by teachers influence their beliefs and expectancies?” (b) “How do undergraduate students’ perceptions of the support they are provided with by their family influence their beliefs and expectancies?” and (c) “How do undergraduate students’ expectancies effect their engagement?”. Specifically, two hypothesized models (M1 and M2) derived from Figure 2 were proposed and tested by the Structural Equation Modeling (SEM) procedure with the Structural Equations Program (EQS) program [72]. The structural configuration and the hypothesized connections addressed in both models are displayed in Figure 3 and Figure 4.

Regarding the first proposed hypothesized model (M1), the following predictions were addressed in the specific Higher Education setting and subject matter contexts: Students’ perceptions of family and teacher support were expected to be good predictors of expectancy–value beliefs (H1); in turn, expectancy–value beliefs were expected to be good predictors of student engagement (H2). 

For the second proposed hypothesized model (M2), the following predictions were addressed: students’ perceptions of teacher and family support were expected to be good predictors of the achievement goals adopted by students (H3); in turn, the achievement goals adopted by students were expected to be good predictors of student engagement (H4). Specifically, students’ perceptions of teacher and family support were expected to have, on the one hand, a positive impact on mastery goals (focused on developing one’s competence), performance goals (focused on demonstrating the competence shown by others), and the self-worth goal (focused on being proud of one’s own competence) and, on the other hand, a negative impact on performance-avoidance goals (focused on avoiding social judgments and humiliation by others) and work-avoidance goals (focused on making the minimum effort to learn).

## 2. Method

### 2.1. Participants and Procedure

Eliminating those students who did not complete all the questionnaires at the three stipulated times left the sample with 267 participants, of whom 39 were male (14.6 %), and 228 were female (85.4%). They were aged between 19 and 48 years. (M = 22.45, SD = 3.62). The participants studied the Educational Psychology and Education degree in academic year 2018–19 at two universities in eastern Spain.

Participation in the study was completely voluntary. Confidentiality and personal data protection were guaranteed in accordance with current Spanish law. 

### 2.2. Measures

Most of the scales employed to measure the variables herein considered were constructed by reviewing and refining the original scales used in previous studies conducted in the university context [7,8,9,73]. The scales about teacher and family support were handed out at time point 1 (at the beginning of the course, after three weeks of class). Those referring to expectancy–value beliefs and achievement goals were handed out at time point 2 (halfway through the course) and the student engagement scale was handed out about two weeks before the course ended (time point 3). The used scales are listed below. See Table 1 for item examples.

**Family support scale.** This scale comprises 12 items and was built ad hoc to measure students’ perception of their family support provided during the course. Students indicated their level of agreement on a Likert response scale ranging from 1 (Totally disagree) to 6 (Totally agree). 

Firstly, an exploratory factor analysis (principal component method with varimax rotation) was conducted on the whole scale composed of 12 items. Three factors (Study support, Affective support, Rules at home) were extracted. They accounted for 77.28% of the total variance. Cronbach’s alpha values ranged between 0.94 (maximum) and 0.77 (minimum).

The extracted factors were used to carry out a confirmatory factor analysis (CFA). The CFA was conducted with the EQS program [72]. The fit index values obtained by the maximum likelihood (ML) method of estimation (Chi-Square = 161.968; D.F. = 51; NNFI = 0.940; CFI = 0.912; RMSEA = 0.090) and the ML robust method of estimation (Chi-Square = 104.802; D.F. = 51; NNFI = 0.949; CFI = 0.961; RMSEA = 0.063) revealed that the model fit the data.

**Teacher support scale.** This is a short version of the original scale validated in the Spanish higher education [10]. This scale contains 21 items and was designed to measure students’ perception of teacher support provided during the students’ learning process. Students indicated their level of agreement on a Likert response scale ranging from 1 (Totally disagree) to 6 (Totally agree).

An exploratory factor analysis (principal component method with varimax rotation) was firstly conducted on the whole scale, which comprised 21 items. Four factors (Motivational support, Formative evaluation support, Relational support, Competence support) were extracted. They accounted for 69.36% of the total variance. Cronbach’s alpha values ranged between 0.91 (maximum) and 0.75 (minimum).

The extracted teacher support factors were used to carry out a CFA. The CFA was performed with the EQS program [72]. The fit index values obtained by the ML method of estimation (Chi-Square = 473.840; D.F. = 183; NNFI = 0.925; CFI = 0.934; RMSEA = 0.072) and the ML robust method of estimation (Chi-Square = 373.735; D.F. = 183; NNFI = 0.921; CFI = 0.931; RMSEA = 0.059) showed that the model fit the data.

**Expectancy–value scale.** This is a short version of the scale devised by Doménech-Betoret, et al. [19]. This 21-item scale was designed to measure undergraduate students’ expectancy–value beliefs of the T–L process conducted for a specific subject. It was structured according to the Motivational Theory proposed by authors in this tradition. For the subject value subscale, students indicated their level of agreement on a 6-point Likert scale, ranging from 1 (Do not agree very much) to 6 (Totally agree). For the expectancy subscales, students indicated their level of agreement on a 6-point Likert scale ranging from 1 (Totally disagree) to 6 (Totally agree).

An exploratory factor analysis (principal component method with varimax rotation) was run on the whole scale with 21 items. Four factors (Subject value, Success expectancy, Enjoyment expectancy with teacher, and Controllability expectancy) were extracted and accounted for 72.40% of the total variance. Cronbach’s alpha values ranged between 0.94 (maximum) and 0.82 (minimum). 

The extracted expectancy–value factors were employed to carry out a CFA, conducted with the EQS program [72]. The fit index values obtained by the ML method of estimation (Chi-Square = 359.511; D.F. = 183; NNFI = 0.512; CFI = 0.957; RMSEA = 0.060) and the ML robust method of estimation (Chi-Square = 302.081; D.F. = 183; NNFI = 0.957; CFI = 0.963; RMSEA = 0.049) indicated that the model satisfactorily fit the data.

**Achievement goals scale.** This scale comes from Doménech-Betoret, et al. [19]. It contains 22 items and was designed to measure the learning goal that undergraduate students adopt during the T–L process. Students indicated their level of agreement with the scale items from 1 (Totally disagree) to 6 (Totally agree).

An exploratory factor analysis (principal component method with varimax rotation) was carried out on the whole scale with 22 items. Five factors (Domain, Performance, Performance-avoidance, Self-worth, Effort avoidance) were extracted and accounted for 82.04% of the total variance. Cronbach’s alpha values ranged between 0.96 (maximum) and 0.70 (minimum). 

The factors extracted from the achievement scale were used to carry out a CFA, which was run with the EQS program [72]. The fit index values obtained by the ML method of estimation (Chi-Square = 502.675; D.F. = 199; NNFI = 0.940; CFI = 0.948; RMSEA = 0.076) and the ML robust method of estimation (Chi-Square = 391.589; D.F. = 199; NNFI = 0.954; CFI = 0.960; RMSEA = 0.060) demonstrated that the model fit the data.

**Behavioral engagement**. This 15-item scale was built ad hoc to assess the degree of student engagement in the T–L process. Students indicated their level of agreement with the scale items from 1 (Totally disagree) to 6 (Totally agree).

An exploratory factor analysis (principal component method with varimax rotation) was run with the whole 15-item scale. Three factors (Attention, Persistence, Dedication) were extracted and accounted for 76.46% of the total variance. Cronbach’s alpha values ranged from 0.94 (maximum) to 0.90 (minimum). 

The factors extracted from the engagement scale were used to carry out a CFA, which was carried out with the EQS program [72]. The fit index values obtained using the ML method of estimation (Chi-Square = 310.427; D.F. = 87; NNFI = 0.919; CFI = 0.933; RMSEA = 0.089) and the ML robust method of estimation (Chi-Square = 210.516; D.F. = 87; NNFI = 0.938; CFI = 0.948; RMSEA = 0.074) revealed that the model satisfactorily fit the data.

### 2.3. Statistical Analyses

The hypothesized connections were tested by structural equation modeling (SEM). The ML and ML robust methods of estimation (if the assumption of multivariate normal distribution was not met), developed by Satorra & Bentler [74], were used, along with the EQS program [72], to calculate the fit indices of the hypothesized models. Given that the Chi-square test is sensitive to sample size, using relative fit indices such as CFI, the NNFI, and RMSEA is highly recommended [75]. Based on a general consensus among authors, NNFI and CFI values above 0.90 [76], or even 0.95 [77], were set as the cut-off point. Values for RMSEA below 0.05 indicate a good fit, whereas values up to 0.08 denote an unacceptable fit [78].

## 3. Results

### 3.1. Descriptive Statistics and Internal Consistency of Scales 

The mean, standard deviation, reliability, and structure of the scales are provided in Table 1. The factor and confirmatory analyses confirmed the scales’ original structure and configuration. Cronbach’s alpha coefficients indicated good internal consistency for all the scales within the 0.70–0.96 range. A construct measure was obtained by averaging the items included in each factor of each scale. See Table 1 for more details. 

### 3.2. Correlation between Variables

A bivariate correlational analysis was carried out as an approach to explore the relation between the variables considered in this study. Firstly, the relations among the provided student support (teacher/family), expectancy–value constructs, and behavioral engagement were explored. The results are shown in Table 2. General positive and significant correlations were obtained between teacher and family support and students’ expectancy–value beliefs (the most remarkable ones were found between teacher support and expectancy–value beliefs) and also between students’ expectancy–value beliefs and student behavioral engagement. See Table 2 for details.

Secondly, the relations among the provided student support (teacher and family), achievement goals and behavioral engagement were explored. The results are shown in Table 3. In general, positive and significant correlations were obtained between teacher–family support and “positive” achievement goals (mastery goal, performance goal, and self-worth goal). The most remarkable ones were found between teacher support and “positive” achievement goals. Positive and significant correlations were also obtained between “positive” achievement goals and student behavioral engagement. Conversely, negative and significant correlations were generally observed between teacher–family support and avoidance achievement goals (performance-avoidance and effort avoidance), and also between avoidance achievement goals and student behavioral engagement. See Table 3 for details.

### 3.3. Structural Equation Modeling

The first hypothesized model (M1) was tested and was optimized when a covariance between two variable errors (E36 persistance–E37 dedication) from the engagement latent variable was introduced, following the recommendations of the Wald and Lagrange test in the EQS program. Then, the model was tested again. Given that the multivariate kurtosis (Mardias’s coefficient = 40.978, normalized estimate = 15.817) indicated that normal distribution was not met, the Maximum Likelihood (ML) robust method of estimation was used. The obtained fit indices (*χ^2^* = 191.341; *p* = 0.0000, d.f. = 73; NNFI = 0.885; CFI = 0.908; RMSEA = 0.078) showed the model’s acceptable data fit. According to the results, teacher–family support had a positive and significant effect on expectancy–value beliefs. In turn, expectancy–value beliefs had a positive and significant effect on student behavioral engagement. See Figure 5 for more details.

The second hypothesized model (M2) was tested. The model was optimized when the avoidance goal’s latent variable was removed and when a covariance between two variable errors (E36 persistence–E37 dedication) from the engagement latent variable was introduced, following the recommendations of the Wald and Lagrange test in the EQS program Then, the model was retested. Given that the multivariate kurtosis (Mardias’s coefficient = 32.273, normalized estimate = 13.351) indicated that normal distribution was not met, the ML robust method of estimation was employed. The obtained fit indices (*χ*^2^ = 138.816; *p* = 0.0000, d.f. = 61; NNFI = 0.907; CFI = 0.927; RMSEA = 0.069) showed the model’s good data fit. According to the results, teacher–family support had a positive and significant effect on “positive” achievement goals (mastery–performance–self-worth goals). In turn, “positive” goals (mastery–performance–self-worth goals) had a positive and significant effect on student behavioral engagement. See Figure 6 for details.

## 4. Discussion

The main objective of this research work was to examine the relations among students’ external support (teacher–family), relevant motivational variables (expectancy–value beliefs and achievement goals), and behavioral engagement. The intention was to provide information that helps to explain why some students are more motivated than others to learn a specific curricular subject in university education. Although the motivation theme has been extensively studied, this research, using the MOCSE model [7,8,9,10] as a framework, examined this complex construct from a new perspective for two main reasons: First, this model, based on the temporal conception of motivation [11,12,13,14,15,16,17,18,19,20,21,22,23,24,25,26,27,28,29,30,31,32,33,34,35,36,37,38,39,40,41,42,43,44,45,46,47,48,49,50,51,52,53,54,55,56,57,58,59,60,61,62,63,64,65,66,67,68,69,70,71,72,73,74,75,76,77,78,79] distinguishes between predecisional and postdecisional motivation, which enables a profound study of motivation. That is, it allows not only the aspect referred to the desire or intention that the student has to learn to be examined but also if that intention is implemented through concrete actions (volition or executive motivation) during the learning process. Note that executive motivation is closely related to [80]. Second, grounded in the socio-cognitive perspective of motivation, expectancy–value and achievement goal theories are, according to the literature, two of the most used theories to study achievement motivation in education. However, the MOCSE model introduces new constructs derived from the job demands–resources model, traditionally used in the workplace, as an attempt to better understand motivation and engagement in the educational setting.

In accordance with the M1 model, teacher and family support had a positive and significant effect on expectancy–value beliefs. In turn, expectancy–value beliefs had a positive and significant effect on student behavioral engagement.

In with the M2 model, teacher–family support had a positive and significant effect on “positive” achievement goals (mastery–performance–self-worth goals). In turn, “positive” goals (mastery–performance–self-worth goals) had a positive and significant effect on student behavioral engagement.

Therefore, the positive effect of teacher–family support on both the considered motivational variable types (expectancy beliefs and adaptive goals) and, in turn, their effect on university student engagement conduct are confirmed.

In our study, the impact of teacher support on the motivational variables (expectancy and learning goals) was stronger than family support. These results are coherent with previous research [20,81,82], which show that teacher effects are stronger than parental effects at lower levels of education than university education. Furrer, Skinner [81] reported that teacher participation better predicts students’ participation and emotional functioning in the classroom than parents’ influence. Marchant, Paulson, Rothlisberg [82] also revealed this teacher support superiority and found that students’ perceptions of their teachers’ emotional support were positively related to their perceived academic competence, values, and interest in what is academic.

In line with these teacher support results, the affective teacher support (motivational and relational support) offered in university education was found to be more closely related to student motivation than instructional support (formative evaluation and competence support) for both expectancies and goals. These findings are coherent with the evidence provided by [57] for the relevance of emotional teacher support for quality teaching and also for student motivation and engagement. Therefore, beliefs in university teachers only having to offer instructional support are contradicted given the importance that relational and affective support has been shown to have on university student motivation.

Our family support results confirmed that this support also had a significant influence on university student motivation, unlike those beliefs which consider that the role of this support is no longer relevant at this level of education. These results are coherent with the findings of studies conducted at lower levels of education [25,26,27,28,29,30,31,32,33,34,35,36,37,38,39,40,41,42,43,44,45,46,47,48,49,50,51,52,53,54,55].

In the present work, affective teacher–family support was positively related to both university student motivation and engagement conducts, as previous research has demonstrated. Following other authors’ proposals [20,26], the “Rules at home” dimension was added to family support. As in former studies (see [46]), setting limits in the family context was corroborated to be significantly related to university student behavioral engagement. It would appear that these limits help university students to positively and adequately behave socially in the education setting.

This study also confirmed the significant effect that expectancy–value beliefs had on behavioral engagement, which falls in line with the results obtained by former works conducted at the university level of education [61,62]. In the same vein, the significant influence that achievement goals had on university students’ behavioral engagement was corroborated, which several studies also report [70,71].

In our study, the goals considered to be “adaptive or positive” (mastery, performance, and self-worth goals) had a positive effect on student engagement. Mastery goals had a more positive and stronger impact on engagement, which falls in line with most research that reached the same conclusion: Mastery goals are the most adaptive type of goal for performance and other results such as engagement [20,21,22,23,24,25,26,27,28,29,30,31,32,33,34,35,36,37,38,39,40,41,42,43,44,45,46,47,48,49,50,51,52,53,54,55,56,57,58,59,60,61,62,63,64,65,66,67,68,69,70,71,72,73,74,75,76,77,78,79,80,81,82,83]. Self-worth goals, which were herein included, also significantly influenced student engagement conducts and even proved somewhat more relevant than performance goals.

Performance goals also positively influenced engagement, albeit to a lesser extent than mastery goals. These findings coincide with those that many research works provide, which indicates that performance goals are a good predictor of all types of performance outcomes, including engagement (see [83]). The majority of these research works were conducted in western educational contexts, which greatly encourage competitiveness, unlike eastern societies that do not and where performance goals are not outstanding predictors of achievements (see [20]).

According to our data, however, no significant impact was found for the performance-avoidance and work-avoidance goals on university student behavioral engagement. The fact that performance-avoidance goals did not significantly influence behavioral engagement can be taken as a logical result in western samples because behavioral engagement is adaptive, and performance-avoidance (attempting to avoid poor results) in our context is not considered adaptive because more importance is attached to obtain positive results. In eastern samples, which are more collectivist, performance-avoidance goals are more normative and are not considered so detrimental. These cultures may value avoiding negative results (see [20]).

The impact that performance-avoidance goals had on the results might be moderated by culture but, as King & McInerney [20] pointed out, the negative impact of performance-avoidance goals seems to be culturally universal and appears in studies carried out in western and eastern cultures. The same authors pointed out that their study found significantly negative correlations between performance-avoidance goals and behavioral engagement for Secondary Education students, which confirmed previous research findings. In our study, performance-avoidance goals also correlated negatively and significantly with engagement. These goals seem to have had this weakening effect on engagement, which has also been found in previous studies performed with Secondary Education students. However, as these goals did not enter the model that fitted the obtained data, it cannot be stated that they had a relevant effect on university student engagement. Analyzing this relation in-depth in future works is recommended.

### 4.1. Conclusions 

Based on MOCSE postulates, two hypothesized models (M1 and M2) were tested by SEM procedure, and the following conclusions were drawn: (a) In accordance with the M1 model, teacher and family support had a positive and significant effect on expectancy–value beliefs. In turn, expectancy–value beliefs had a positive and significant effect on student behavioral engagement. The results are going in the expected direction; (b) In accordance with the M2 model, teacher–family support had a positive and significant effect on “positive” achievement goals (mastery–performance–self-worth goals). In turn, “positive” goals (mastery–performance–self-worth goals) had a positive and significant effect on student behavioral engagement. The results are going in the expected direction; (c) Important educational implications can be derived from these findings to improve students’ behavioral engagement in the university context. In short, the present study sheds light on how teachers and families can influence university student motivation and engagement. Our work stresses the impact of both support types on developing adaptive learning goals and on student expectancy–value beliefs in the classroom context, which, in turn, influences behavioral engagement.

### 4.2. Practical Implications

Practical implications for university teachers who attempt to motivate students to engage more in their learning are derived from our study. By means of teaching styles that facilitate support (with motivational and relational support behaviors and with classic teacher support), teachers can improve students’ beliefs and goals so they can, in turn, encourage students to manifest engagement behavior. As some authors have pointed out [84,85], working on students’ beliefs with teacher support can act as a promising approach to improve student motivation and learning.

Our results also suggest that it is important to inform university students that although counting on family support (help with studying, affective support, a well-structured environment, rules) is less relevant than counting on teacher support, this can still help students to improve their motivation and, consequently, their engagement to study.

### 4.3. Limitations and Suggestions for Future Research

Although the results obtained herein are satisfactory, some limitations and suggestions for future research should be pointed out.

The sample herein employed implied two frequent problems (limitations) in the Social Sciences area: it does not meet the multivariate normality assumption, and the sample size was small. However, the ML robust method of estimation developed by Satorra & Bentler [74,75] was followed, which is suitable when both the above problems come into play [79,80]. In any case, further research is recommended to analyze if the obtained results can be validated for larger samples.

Self-reported measures were also employed to evaluate the variables selected in our study. It is important to carry out studies in the future that consider using teacher reports to supplement students’ self-reported measures.

Our study results corroborate that family support dimensions (academic, family, and rules at home) influence academic motivation. If other research works about family support (see [26]) were contemplated, the academic and family dimensions correspond to participation, whereas rules at home form part of the structure. However, these research works add another dimension that is related to motivation and academic achievement which the present work did not include: Autonomy support (e.g., “My parents encourage me to see how what I learn can be useful for me”). This dimension should be included in future works because some studies have demonstrated the positive impact that parents’ autonomy support has on student motivation [86].

Nor did our study consider the role of autonomy support offered by teachers to their students. Former research based on [87,88] the SDT model of basic needs has provided a considerable body of evidence about how psychological needs (competence, relation, autonomy) directly impact well-being and motivation [89]. Our work checked how competence-related teacher (instructional) support and the (affective support) relation positively influence academic motivation. Exploring how autonomy support can facilitate teachers is a pending future research point.

For performance-avoidance goals, our study found a negative significant relation that previous research has indicated between these goals and student engagement, even though these goals did not enter the model with the obtained data. As most authors (see [20]) stress the strong negative impact that these goals have on student engagement/achievement, this relation should be explored in-depth in future works.

Only the unidirectional effects of goals and expectancies on the engagement results were examined, although engagement behaviors may influence expectancies and the goal types that students pursue. Although our data do not allow us to prove this proposal, future studies can be conducted to perform these analyses.

The relation of learning goals and other motivational variables to other learning outcomes, and not only to engagement, should also be analyzed. For instance, it would also be interesting to study in the future the impact of the goals on student well-being [20] since there is little evidence in this regard.

Finally, the present work does not consider the role of students’ culture. Most studies about student participation have been carried out in western societies [40]. However, some research works [47,90] show a differential effect of perceived teacher–family support on student engagement according to their culture. Therefore, it would be interesting to perform this differential analysis in future works.

## Figures and Tables

**Figure 1 ijerph-18-02606-f001:**
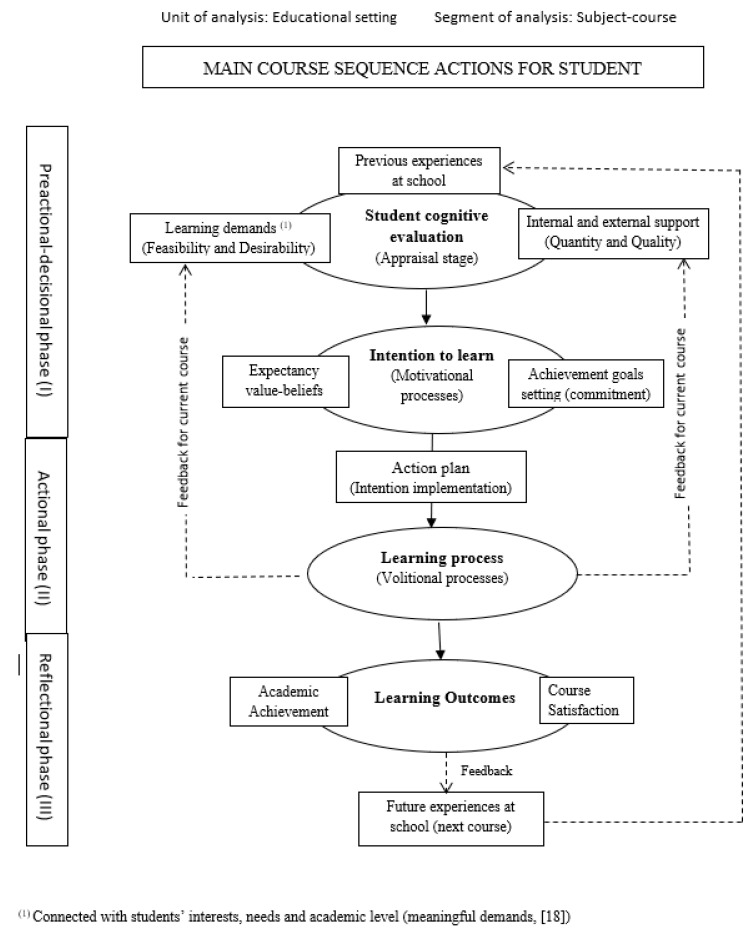
MOCSE diagram: the main course sequence actions for students.

**Figure 2 ijerph-18-02606-f002:**
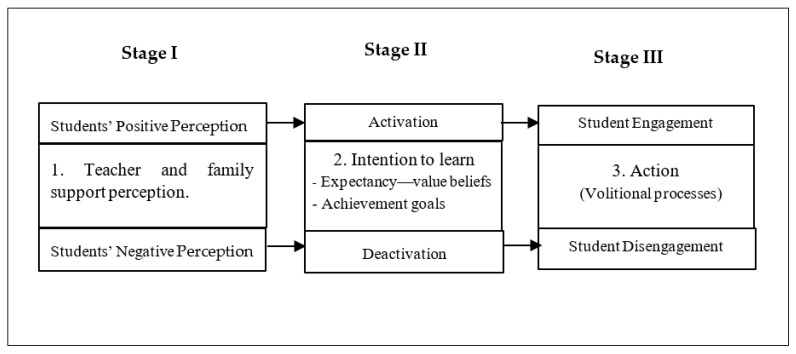
Adaptation of the MOCSE general framework to study student engagement.

**Figure 3 ijerph-18-02606-f003:**
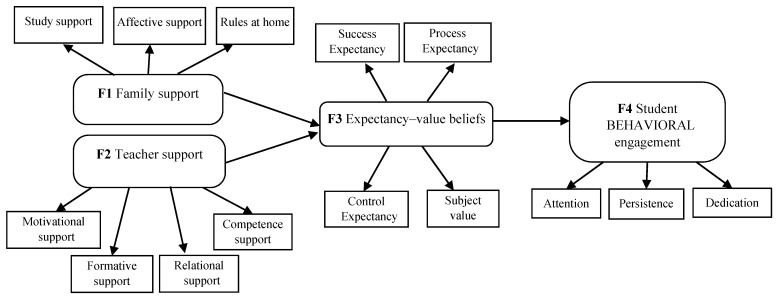
Hypothesized model (M1). Relations among family (Factor 1) and teacher (Factor 2) support, expectancy–value beliefs (Factor 3), and student behavioral engagement (Factor 4).

**Figure 4 ijerph-18-02606-f004:**
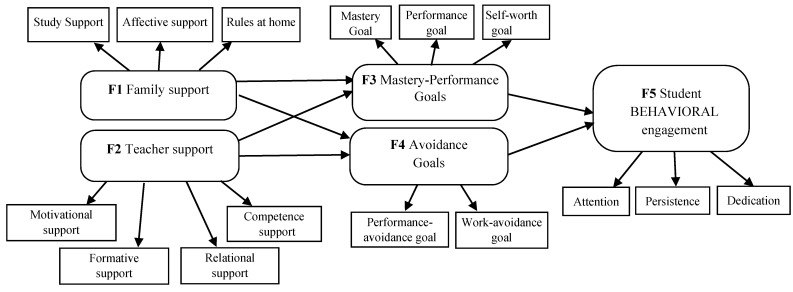
Hypothesized model (M2). Relation among family (Factor 1) and teacher (Factor 2) support, achievement goals (Factor 3, Factor 4), and student behavioral engagement (Factor 5).

**Figure 5 ijerph-18-02606-f005:**
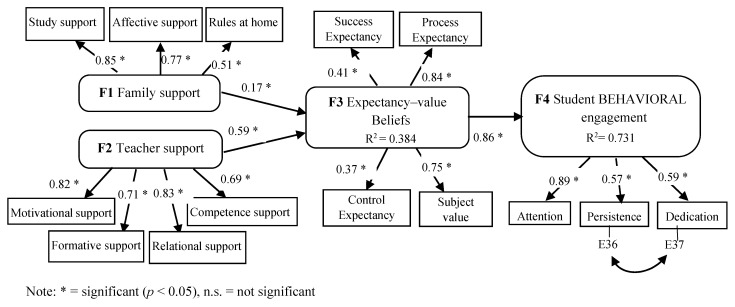
Optimized model (M1). Relation among teacher–family support, expectancy–value beliefs, and student behavioral engagement. The structural configuration and standardized coefficients of the optimized model are displayed.

**Figure 6 ijerph-18-02606-f006:**
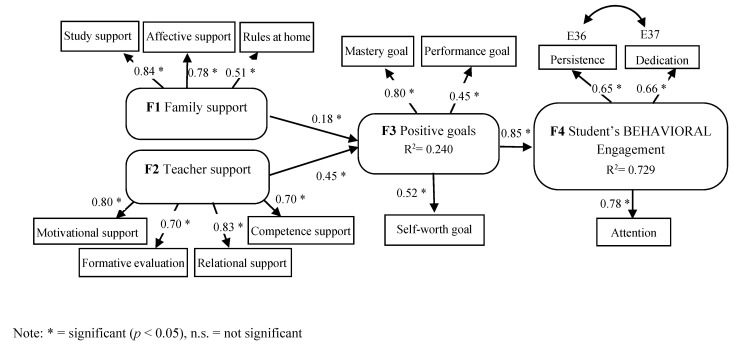
Optimized model (M2). Relation among teacher–family support, achievement goals, and student behavioral engagement. The structural configuration and standardized coefficients of the optimized model are displayed.

**Table 1 ijerph-18-02606-t001:** Summary of the exploratory factor analysis and internal consistency of scales.

Scales	Factors	Items (n)	M	S.D.	Variance	Cronbach’s α
(Minimum = 1; Maximum = 6)						
*Family support*	3	12			77.28	
F1: Affective support		5	4.92	1.20	35.79	0.94
F2: Study support		3	4.84	1.24	21.44	0.90
F3: Rules at home		4	4.74	1.07	20.04	0.77
*Teacher support*	4	21			69.36	
F1: Motivational support		6	4.20	0.99	20.69	0.91
F2: Relational support		6	4.94	0.78	18.95	0.90
F3: Self-competence support		5	4.80	0.85	18.15	0.91
F4: Formative evaluation (teacher feedback)		4	4.52	0.76	11.57	0.75
*Expectancy–value beliefs*	4	21			72.40	
F1: Success expectancy		10	4.17	0.76	30.75	0.94
F2: Subject value		4	3.34	1.01	15.02	0.82
F3: Control expectancy		4	5.00	0.75	14.52	0.87
F4: Process expectancy: Feeling good		3	3.96	1.19	12.11	0.93
*Achievement goals*	5	22			82.04	
F1: Performance-Avoiding goal		5	2.02	1.26	20.22	0.96
F2: Mastery goal		5	4.95	1.02	20.23	0.95
F3: Performance goal		5	3.12	1.35	19.10	0.96
F4: Avoiding effort goal		4	2.25	1.05	13.62	0.88
F5: Self-worth goal		3	4.17	0.98	8.56	0.70
*Behavioral engagement*	3	15			76.46	
F1: Attention		6	4.58	0.90	28.02	0.92
F2: Persistence		5	3.71	1.23	27.08	0.94
F3: Dedication		4	4.20	1.04	21.36	0.90
*Family support* F1: Affective support: “Mis padres me transmiten constantemente su afecto y cariño” [“My parents always make known their affection and love for me”] F2: Study support: “Mis padres suelen preguntarme por mis estudios” [“My parents tend to ask me about my studies”] F3: Rules at home: “En mi casa no existen las normas, cada uno hace lo que quiere y cuando quiere” [“We have no rules at home. Everyone does whatever they want, when they want”] (Reversed code) *Teacher support* F1: Motivational support: “Desde el principio, el profesor se esforzó por despertar nuestra curiosidad e interés por esta materia” [“From the beginning, the teacher made an effort to awaken our curiosity and interest in this subject”] F2: Relational support: “Por lo que he visto estos primeros días de clase, creo que el profesor/a será una persona cercana” [“From what I have seen on these first days of class, I think the teacher will be close”] F3: Self-competence support: “Desde el principio, el profesor nos ha trasmitido la idea de que todos estamos capacitsdos para superar esta materia si nos lo proponemos” [“From the beginning, the teacher has conveyed us the idea that we are all qualifyied to overcome this matter if we propose it”] F4: Formative evaluation (teacher feedback): “El sistema de evaluación torga mucha importancia al trabajo continuado del estudiante y al feedback del profesor” [“The evaluation system attaches much importance to students’ continued work and the teacher’s feedback”] *Expectancy–value beliefs* F1: Success expectancy: “¿Piensas que serás capaz de obtener buenas notas en esta materia?” [“Do you think you will be able to obtain good marks for this subject?”] F2: Subject value: “¿Cómo es de útil esta materia para tí?” [“How useful is this subject for you?”] (usefulness) F3: Control expectancy: “¿En qué medida crees que influirá tu dedicación a la asignatura en tu nota final?” [“To what extent do you think that your dedication to the subject will influence your final mark?”] F4: Process expectancy: Feeling good: “¿Crees que te sentirás bien en clase con este profesor/a, de aquí a final de curso?” [“Do you think you will feel comfortable with this teacher from this point onwards until the end of the course?”] *Achievement goals* F1: Performance-avoidance goal: “Mi objetivo en esta asignatura es evitar que mis compañeros y el profesor/a piensen que soy un tonto” [“My goal in this subject is to avoid my classmates and my teacher thinking I’m a fool”] F2: Mastery goal: “Mi objetivo en esta asignatura es aprender todo lo que pueda” [“My goal in this subject is to learn as much as possible”] F3: Performance goal: “Mi objetivo en esta asignatura es demostrar a mis compañeros/as y al profesor/a que soy bueno en esta materia” [“My goal in this subject is to show my classmates and my teacher that I’m good in this subject”] F4: Effort-avoidance goal: “Mi objetivo en esta asignatura es superarla con el mínimo esfuerzo” [“My goal in this subject is to pass it by making the minimum effort”]. F5: Self-worth goal: “Mi objetivo en esta asignatura es experimentar el orgullo que sigue al éxito” [“My goal in this subject is to feel the pride that comes with success”] *Behavioral engagement* F1: Persistence: “Cuando me enfrentaba a una tarea o reto difícil, trataba de esforzarme más” [“I’ve tried to make more effort with difficult tasks or challenges”] F2: Attention: “He seguido con atención e interés las explicaciones del profesor/a” [“I’ve paid attention to and shown interest in the teacher’s explanations”] F3: Dedication: “El tiempo y esfuerzo que he dedicado a esta materia ha sido el adecuado para su comprensión y dominio” [“The time and effort I’ve spent on this subject have allowed me to understand and master it”]						

**Table 2 ijerph-18-02606-t002:** Pearson’s bivariate correlations between the constructs considered in M1.

	1	2	3	4	5	6	7	8	9	10	11	12	13	14	15	16
1. Gender	1															
2. Age	0.001	1														
3. Motivation sup.	−0.021	0.051	1													
4. Feedback	0.057	−0.023	0.581 **	1												
5. Relation sup.	0.099	0.035	0.662 **	0.598 **	1											
6. Competence sup.	−0.004	0.042	0.555 **	0.501 **	0.598 **	1										
7. Study support	0.083	−0.246 **	0.119	0.196 **	0.152 *	0.036	1									
8. Affective support	0.133 *	−0.169 **	0.071	0.161 **	0.148 *	0.091	0.658 **	1								
9. Rules at home	0.146 *	−0.067	−0.004	0.059	0.073	−0.077	0.428 **	0.398 **	1							
10. Subject value	0.044	−0.071	0.401 **	0.267 **	0.358 **	0.163 **	0.229 **	0.181 **	0.187	1						
11. Succ. Expect.	−0.058	−0.084	0.232 **	0.178 **	0.217 **	0.376 **	0.062	0.062	0.090	0.303 **	1					
12. Process Expect.	−0.002	−0.040	0.527 **	0.381 **	0.460 **	0.250 **	0.221 **	0.112	0.134 *	0.646 **	0.320 **	1				
13. Control Expect.	0.090	−0.048	0.213 **	0.233 **	0.282 **	0.168 **	0.064	0.121 *	−0.003	0.319 **	0.241 **	0.274 **	1			
14. Engag. Attent.	−0.050	0.082	0.453 **	0.319 **	0.345 **	0.237 **	0.112	0.084	0.167 **	0.583 **	0.314 **	0.665 **	0.253 **	1		
15. Engag. Persist.	0.058	0.064	0.226 **	0.186 **	0.263 **	0.258 **	0.105	0.041	0.198 **	0.413 **	0.300 **	0.371 **	0.331 **	0.512 **	1	
16. Engag. Dedic.	0.088	−0.001	0.149 *	0.135 *	0.118	0.166 **	0.015	0.079	0.250 **	0.388 **	0.370 **	0.417 **	0.316 **	0.529 **	0.617 **	1

Note: Gender: 1 Male; 2 Female; * *p* < 0.05, ** *p* < 0.01.

**Table 3 ijerph-18-02606-t003:** Pearson’s bivariate correlations between the constructs considered in M2.

	1	2	3	4	5	6	7	8	9	10	11	12	13	14	15	16	17
1.Gender	1																
2. Age	0.001	1															
3. Motivation sup.	−0.021	0.051	1														
4. Feedback	0.057	−0.023	0.581 **	1													
5. Relation sup.	0.099	0.035	0.662 **	0.598 **	1												
6. Competence sup.	−0.004	0.042	0.555 **	0.501 **	0.598 **	1											
7. Study support	0.083	−0.246 **	0.119	0.196 **	0.152 *	0.036	1										
8. Affective support	0.133 *	−0.169 **	0.071	0.161 **	0.148 *	0.091	0.658 **	1									
9. Rules at home	0.146 *	−0.067	−0.004	0.059	0.073	−0.077	0.428 **	0.398 **	1								
10. Mastery goal	0.075	−0.037	0.338 **	0.274 **	0.385 **	0.215 **	0.171 **	0.141 *	0.102	1							
11. Perform goal	0.057	−0.051	0.065	0.082	0.045	0.002	0.169 **	0.151 *	0.217 **	0.366 **	1						
12. Perform_avoid	0.031	−0.018	−0.069	−0.067	−0.170 **	−0.121 *	0.041	0.056	0.138 *	0.098	0.551 **	1					
13. Self-worth	0.128 *	0.053	0.085	0.153 *	0.112	0.100	0.137 *	0.135 *	0.145 *	0.417 **	0.374 **	0.283 **	1				
14. Avoiding	−0.217 **	−0.088	−0.204 **	−0.197 **	−0.301 **	−0.110	−0.146 *	−0.064	−0.222 **	−0.307 **	−0.077	0.138 *	−0.167 **	1			
15. Engag. Attent.	−0.050	0.082	0.453 **	0.319 **	0.345 **	0.237 **	0.112	0.084	0.167 **	0.541 **	0.268 **	0.145 *	0.313 **	−0.177 **	1		
16. Engag. Persist.	0.058	0.064	0.226 **	0.186 **	0.263 **	0.258 **	0.105	0.041	0.198 **	0.459 **	0.209 **	0.083	0.339 **	−0.263 **	0.512 **	1	
17. Engag. Dedic.	0.088	−0.001	0.149 *	0.135 *	0.118	0.166 **	0.015	0.079	0.250 **	0.443 **	0.313 **	0.140 *	0.343 **	−0.207 **	0.529 **	0.617 **	1

Note: Gender: 1 Male; 2 Female; * *p* < 0.05, ** *p* < 0.01.

## Data Availability

Data available on request due to restrictions eg privacy or ethical. The data presented in this study are available on request from the corresponding author. The data are not publicly available due to data protection.

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
