# Peer review of "Influence of Teacher and Family Support on University Student Motivation and Engagement"

_ijerph, 2021, doi:10.3390/ijerph18052606_

Round 1
Reviewer 1 Report
This article's main objective was analyzing the relation between students' support and engagement, while also factoring in various motivational variables.
I have noticed many elements in the article that require improvement.
The authors should utilize a more neutral language in scientific articles. In this case, I had a hard time understanding who the authors are referring to when writing "we"; is it the writers of the article, society, or something else?
The authors should justify why it is important to make use of the model (Figure 1) in this manuscript. The only justification I have currently noticed is Line: 47 "Figure 1 represents the sequence of students’ actions for one course according to the MOCSE postulates". The manuscript could use some more information justifying the use of this figure and describing it fully. The authors should characterize it, and if they decided to keep it, it should be mentioned in the discussion.
I have noticed a few editorial choices in the text which negatively affect the readability of the text. In particular, the method of referencing particular articles without quoting the author is difficult to parse in a few cases, for example:
Line 83: „Nevertheless as [16] point out, parents…’” – I recommend writing "Nevertheless as Spera [16] points out, parents..." This occurs in a few other lines: 110, 113, 114, 145, 197, 207, 209, 2019, 227, 353, 575
When referencing the hypotheses presented visually in Figure 2, they should be written out separately as well. Their current method of integration in the text makes it difficult to coherently follow. This is especially important since the model appears to be missing some words, perhaps due to a formatting issue. (Line 237-256)
In lines 88-89, the claim made requires some further examples. Please provide references.
Line 218: "Very few research works have simultaneously contemplated the influence of perceived support in students’ achievement goals and engagement" – how was this conclusion reached? Have the authors made some sort of meta-analysis? If so, this should be emphasized here. If not, some other research to support the claim should be included.
In line 181-182, the reference is placed in an awkward spot, making it difficult to read. Upon first reference to these works, they should be included, instead of later in the paragraph.
There are many scales included in the measurements; were these scales created by the authors, or someone else? If someone else, what was the text's native language? If the text was in a different language than Spanish, were any cultural adaptations made? If these questions are not relevant, it would be best to include that the scales were made for the study.
Outside of these recommended changes, the article explores an interesting topic, and readable, though another round of proofreading is recommended. The article's dense style could be difficult for non-scientific readers.
Author Response
Comments and Suggestions for Authors
I VERY MUCH APPRECIATE THE EFFORT THAT REVIEWER HAS MADE TO REVIEW THIS ARTICLE, AND ALSO HIS/HER RECOMMENDATIONS THAT WILL IMPROVE THE QUALITY OF THE PAPER.
FOLLOWING REVIEWER 1’S RECOMMENDATIONS, I HAVE MADE ALL THE SUGGESTED CORRECTIONS AND CHANGES
This article's main objective was analyzing the relation between students' support and engagement, while also factoring in various motivational variables.
I have noticed many elements in the article that require improvement.
The authors should utilize a more neutral language in scientific articles. In this case, I had a hard time understanding who the authors are referring to when writing "we"; is it the writers of the article, society, or something else?
FOLLOWING REVIEWER 1’S RECOMMENDATIONS, I HAVE REPLACED “WE” WITH MORE NEUTRAL TERMS IN ALL DE MANUSCRIPT.
The authors should justify why it is important to make use of the model (Figure 1) in this manuscript. The only justification I have currently noticed is Line: 47 "Figure 1 represents the sequence of students’ actions for one course according to the MOCSE postulates". The manuscript could use some more information justifying the use of this figure and describing it fully. The authors should characterize it, and if they decided to keep it, it should be mentioned in the discussion.
Doménech-Betoret, F. (2018) The Educational Situation Quality Model: Recent Advances. Front. Psychol. 9:328. doi: 10.3389/fpsyg.2018.00328
MOCSE IS NOT SIMPLY A THEORETICAL MODEL THAT AIMS TO GUIDE RESEARCH IN THE EDUCATIONAL CONTEXT, LIKE MOST OF THE INSTRUCTIONAL MODELS FOUND IN THE LITERATURE, BUT ALSO PROVIDES A METHODOLOGICAL PROCEDURE FOR TEACHERS TO APPLY IT IN THE CLASSROOM. SO MORE THAN A CONCEPTUAL FRAMEWORK, IT IS ALSO A USEFUL TOOL THAT PROVIDES THE KEYS AND SYSTEMATIC GUIDELINES (EVALUATION, REFLECTION, INTERVENTION) SO THAT TEACHERS ARE ABLE TO DIAGNOSE THE MOTIVATIONAL DEFICIENCIES OF THEIR STUDENTS AND TO IDENTIFY THEIR POSSIBLE CAUSES IN TERMS OF DEMANDS AND SUPPORTS/RESOURCES. IT IS VALUABLE INFORMATION FOR DESIGNING IMPROVEMENT ACTIONS TO CORRECT DETECTED MOTIVATIONAL DEFICIENCIES AND TO IMPROVE ACADEMIC OUTCOMES. IMPROVEMENT ACTIONS CAN BE IMPLEMENTED EITHER DURING THE COURSE UNDERWAY OR THE NEXT COURSE (FOR MORE DETAILS, SEE DOMÉNECH-BETORET, 2018).
ANOTHER SPECIFIC ADVANTAGE OF THE MODEL FOR STUDYING SCHOOL ENGAGEMENT IS THAT MOCSE, BASED ON THE TEMPORAL CONCEPTION OF MOTIVATION (THEORY OF ACTION CONTROL, HECKHAUSEN AND KUHL, 1985; DÖRNYEI, 2000) DISTINGUISHES BETWEEN PREDECISIONAL MOTIVATION (DESIRE, INTENTION) AND POSTDECISIONAL MOTIVATION (EXECUTIVE OR VOLITIONAL MOTIVATION). THIS DISTINCTION ALLOWS PREVENTIVE MEASURES CENTERING ON STUDENTS’ ANTICIPATORY COGNITIVE MOTIVATORS TO BE ADOPTED (E.G., EXPECTATIONS) (FOR DETAILS, SEE DOMÉNECH-BETORET, 2018).
Heckhausen, H., and Kuhl, J. (1985). “From wishes to action: the dead ends and short cuts on the long way to action,” in Goal-directed Behaviour: The Concept of Action in Psychology, eds M. Frese and J. Sabini (Hillsdale, NJ: Lawrence Erlbaum), 134–160.
Dörnyei, Z. (2000). Motivation in action: Towards a process-oriented conceptualization of student motivation. British Journal of Educational Psychology, 70, 519-538.
THE REVIEWER 2’S SUGGESTS PRESENTING AN ADAPTATION OF THE GENERAL FRAMEWORK TO GUIDE THE STUDY OF STUDENTS’ ENGAGEMENT.
FIGURE 1 PRESENTS AN OVERVIEW OF THE GENERAL FRAMEWORK OF THE MODEL CENTERED ON STUDENTS. THIS GENERAL FRAMEWORK WAS ADAPTED TO STUDY STUDENT ENGAGEMENT BASED ON THE MOCSE POSTUATES AND TAKING PREVIOUS PROPOSALS IN THIS FIELD (DOMÉNECH-BETORET, ABELLÁN-ROSELLÓ, 2017; DOMÉNECH-BETORET, GÓMEZ-ARTIGA & ABELLÁN-ROSELLÓ, 2019) AS REFERENCES. PLEASE SEE FIGURE 2.
AUTORES: Fernando Doménech Betoret, Amparo Gómez Artiga y Laura Abellán Roselló
TÍTULO: The Educational Situation Quality Model: A new tool to explain and improve academic achievement and course satisfaction.
REF. REVISTA: Front. Psychol., 18 July 2019 | https://doi.org/10.3389/fpsyg.2019.01692
AUTHORS: Fernando Doménech Betoret y Laura Abellán Roselló
Title: Guía práctica para mejorar la motivación del alumnado de educación secundaria y formación profesional.
REF. TO BOOK: Col.lecció Educació. Publicaciones de la Universitat Jaume I, Castellón, 2017.
Figure 2. Adaptation of the MOCSE general framework to study student engagement.
I have noticed a few editorial choices in the text which negatively affect the readability of the text. In particular, the method of referencing particular articles without quoting the author is difficult to parse in a few cases, for example:
Line 83: „Nevertheless as [16] point out, parents…’” – I recommend writing "Nevertheless as Spera [16] points out, parents..." This occurs in a few other lines: 110, 113, 114, 145, 197, 207, 209, 2019, 227, 353, 575
THANK YOU FOR POINTING THIS OUT. THESE DEFICIENCIES HAVE BEEN CORRECTED.
When referencing the hypotheses presented visually in Figure 2, they should be written out separately as well. Their current method of integration in the text makes it difficult to coherently follow. This is especially important since the model appears to be missing some words, perhaps due to a formatting issue. (Line 237-256)
I AM SORRY, BUT I DO NOT UNDERSTAND WHAT REVIEWER 1 MEANS BY “WHEN REFERENCING THE HYPOTHESES PRESENTED VISUALLY IN FIGURE 2, THEY SHOULD BE WRITTEN OUT SEPARATELY AS WELL”
THE FORMATING PROBLEMS OF FIGURES 2 AND 3 HAVE BEEN CORRECTED
In lines 88-89, the claim made requires some further examples. Please provide references.
EMPIRICAL STUDIES FOCUSING ON FAMILIES HAVE PROVEN THE SIGNIFICANT ROLE OF PARENTS TO ACT AS CONTRIBUTORS TO SCHOOL ENGAGEMENT AND STUDENT PERFORMANCE AT SCHOOL (BEMPECHAT AND SHERNOFF, 2012). FAMILY MEMBERS WHO PROVIDE ACADEMIC (E.G. ASSISTING WITH HOMEWORK) OR MOTIVATIONAL (E.G. RECOGNISING EFFORT AND ENHANCING PROGRESS) SUPPORT HELP TO IMPROVE STUDENTS’ ACADEMIC PERFORMANCE (JELAS, AZMAN, ZULNAIDI AND AHMAD, 2016). MOREOVER, PARENTS’ SUPPORT AND INVOLVEMENT MAY INFLUENCE THE WAY THEIR CHILDREN PERCEIVE THEIR OWN ABILITIES OR THE WAY THEY VALUE SUBJECTS (RODRÍGUEZ, PIÑEIRO, GÓMEZ, REGUEIRO, ESTÉVEZ, AND VALLE, 2017).
Line 218: "Very few research works have simultaneously contemplated the influence of perceived support in students’ achievement goals and engagement" – how was this conclusion reached? Have the authors made some sort of meta-analysis? If so, this should be emphasized here. If not, some other research to support the claim should be included.
THIS SENTENCE HAS BEEN REWRITTEN “AS FAR AS WE KNOW ………”
In line 181-182, the reference is placed in an awkward spot [LUGAR INCÓMODO], making it difficult to read. Upon first reference to these works, they should be included, instead of later in the paragraph.
FOLLOWING REVIEWER 1’S SUGGESTION, THE REFERENCES HAVE BEEN REPLACED AS SO: “THE FINDINGS OF THESE WORKS [18, 37-44] INDICATE…”
There are many scales included in the measurements; were these scales created by the authors, or someone else? If someone else, what was the text's native language? If the text was in a different language than Spanish, were any cultural adaptations made? If these questions are not relevant, it would be best to include that the scales were made for the study.
FOLLOWING REVIEWER 1’S RECOMMENDATION, THE USED SCALES HAVE BEEN INCLUDED IN BILINGUAL SPANISH (ORIGINAL VERSION) AND ENGLISH TO IMPROVE ITS COMPREHENSION FOR ENGLISH SPEAKING PEOPLE.
Outside of these recommended changes, the article explores an interesting topic, and readable, though another round of proofreading is recommended. The article's dense style could be difficult for non-scientific readers.
APART FROM THE COMMENTS MADE HERE IN THE FORUM IN RESPONSE TO THE REVIEWERS’ SUGGESTIONS, I ALSO RESUBMIT THE MANUSCRIPT WITH THE CHANGES MADE HIGHLIGHTED IN GREEN TO FACILITATE THEIR PLACEMENT AND CHECKING.
Reviewer 2 Report
Dear authors, thank you for the opportunity to read your paper the purpose of which, according to my understanding, is to examine the effects of external teacher and family support on student’s motivational aspects and student engagement.
The paper was well written, and in most parts, it was easy to follow the logic of argumentation. The purpose of the study, research design and methodology, and the findings are presented clearly and concisely. All in all, I think the paper in general presents a well-conducted and reported study.
I, however, have some concerns regarding the contribution of the study, and they can be seen as either minor or major, depending on the way these questions are to be answered. I’m somewhat worried about the justification and therefore also the ‘so what’ question regarding the interest of this study. So I would like to point out some need for better justifications for the choices made in the introduction as well as returning to the identified research gap also in the discussion section.
- Introduction
- This section is really dense, in the sense that even though what is being done is basically explained here, I had to read the introduction many times and still was not sure if I understood the choices you have made regarding the framework and the constructs. So I would like to suggest this part be rewritten in a way that it would better provide justifications for the choices made:
- Research gap. You state on the first page that “Although a large body of literature on this field exists, very few research works have analysed the relation between students’ support and engagement, and the influence of other mediator variables on this relation.” I would like to see references to this large body of literature and this identified research gap. Also, you state that “The present study considers this analysis from a new perspective by using the Educational Situation Quality Model (MOCSE, its acronym in Spanish) of Doménech-Betoret, [1-4] as a reference framework.” Could you be more specific on what you mean with the new perspective? How does the approach you take differ from the existing body of knowledge, how does this selected perspective add to that body? So I would, however, like to see more justifications for your research aim in the beginning, as this relates to the way you succeed in answering the ‘so what’ question regarding the research. I agree with the importance of understanding the link between support and engagement but would like to see the research gap presented in light of the existing body of knowledge.
- Utilizing MOCSE framework. I believe I understand roughly what this framework is, but I would like to see the reasoning for adopting this. What is its added value when taking into consideration that you state your constructs are derived from other sources (Job Demands-Resources Model (JD-R) [5], the Expectancy-Value Theory [6], and the Achievement Goal Theory [7-10].)? How are you utilizing MOCSE in particular? My point for asking this relates to the presentation of Figure 1. My view is that the figure should present the framework of _your study_, and not the MOCSE framework since now there are a lot of concepts, relationships, and phases in the figure that is not in the focus of this study at hand, and also not being explained in the text at all. The figure should not have to explain itself, but all the central elements of this study should be opened in the text. A figure is a good way to present an overview of the framework of the study, but it should be then about this particular study at hand. Perhaps you could just refer to MOCSE and state that you present an adaptation of this model, and then focus on explaining how you do this. So I would suggest developing Section 1 of the paper in a way that would better address this particular study at hand and focus on its concepts.
Discussion
- The problem regarding the ‘so what’ question I presented above now realizes in this section, as you state on p. 14 that “Although the motivation theme has been well-studied, with the present study we contribute a new approach using the MOCSE model [1-4] as a reference framework.” – This statement does not really tell anything about your contribution, it merely states that you have adopted a ‘new approach’. In what way is it novel (if you argue it is new, you need to show this in the light of the existing literature); and how does this new approach improve our understanding of student motivation/the research problem you have posed? I.e., ‘so what’?
Methods:
- On page 8 you state that “Those who did not complete all the questionnaires were eliminated, which considerably reduced the original sample size.” So how many respondents did you finally have?
I hope you will find these observations helpful when developing your study further. Thank you once again for the opportunity to read your paper and wishing you all the best with your work!
Author Response
Comments and Suggestions for Authors
Dear authors, thank you for the opportunity to read your paper the purpose of which, according to my understanding, is to examine the effects of external teacher and family support on student’s motivational aspects and student engagement.
The paper was well written, and in most parts, it was easy to follow the logic of argumentation. The purpose of the study, research design and methodology, and the findings are presented clearly and concisely. All in all, I think the paper in general presents a well-conducted and reported study.
I, however, have some concerns regarding the contribution of the study, and they can be seen as either minor or major, depending on the way these questions are to be answered. I’m somewhat worried about the justification and therefore also the ‘so what’ question regarding the interest of this study. So I would like to point out some need for better justifications for the choices made in the introduction as well as returning to the identified research gap also in the discussion section.
I VERY MUCH APPRECIATE THE EFFORT THAT REVIEWER HAS MADE TO REVIEW THIS ARTICLE, AND ALSO HIS/HER RECOMMENDATIONS THAT WILL IMPROVE THE QUALITY OF THE PAPER.
FOLLOWING REVIEWER 2’S RECOMMENDATIONS, I HAVE MADE ALL THE SUGGESTED CORRECTIONS AND CHANGES.
Introduction
This section is really dense, in the sense that even though what is being done is basically explained here, I had to read the introduction many times and still was not sure if I understood the choices you have made regarding the framework and the constructs. So I would like to suggest this part be rewritten in a way that it would better provide justifications for the choices made:
Research gap. You state on the first page that “Although a large body of literature on this field exists, very few research works have analysed the relation between students’ support and engagement, and the influence of other mediator variables on this relation.” I would like to see references to this large body of literature and this identified research gap.
- REGARDING TEACHING SUPPORT
MOST AUTHORS HAVE USUALLY DISTINGUISHED BETWEEN INSTRUCTIONAL-INSTRUMENTAL AND EMOTIONAL-SOCIAL SUPPORT. A NUMBER OF STUDIES HAVE CENTRED ON TEACHERS AS BEING INSTRUCTIONAL SUPPORTIVE AND FOUND INSTRUCTIONAL SUPPORT TO BE RELATED TO SOCIAL, BEHAVIOURAL AND ACADEMIC OUTCOMES (MALEKI AND DEMARAY, 2003). HOWEVER, MANY STUDIES HAVE ALSO CENTRED ON TEACHERS BEING EMOTIONALLY SUPPORTIVE AND FOUND EMOTIONAL SUPPORT TO BE RELATED TO HIGH LEVELS OF INTRINSIC MOTIVATION (KATZ, KAPLAN, AND GUETA, 2009), INTEREST (WENTZEL, BATTLE, RUSSELL, AND LOONEY, 2010) AND ACADEMIC EFFORT (REYES, BRACKETT, RIVERS, WHITE, AND SALOVEY, 2012).
- REGARDING FAMILY SUPPORT
SUPPORT FOR STUDENTS CAN ALSO BE PROVIDED BEYOND THE CLASSROOM CONTEXT. EMPIRICAL STUDIES THAT HAVE FOCUSED ON FAMILIES HAVE PROVEN THE SIGNIFICANT ROLE OF PARENTS TO ACT AS CONTRIBUTORS TO SCHOOL ENGAGEMENT AND STUDENT PERFORMANCE AT SCHOOL (BEMPECHAT AND SHERNOFF, 2012). FAMILY MEMBERS WHO PROVIDE ACADEMIC (E.G. ASSISTING WITH HOMEWORK) OR MOTIVATIONAL (E.G. RECOGNISING EFFORT AND ENHANCING PROGRESS) SUPPORT HELP TO IMPROVE STUDENTS’ ACADEMIC PERFORMANCE (JELAS, AZMAN, ZULNAIDI AND AHMAD, 2016).
ALTHOUGH A LARGE BODY OF LITERATURE IN THIS FIELD EXISTS, SCARCE RESEARCH HAS ANALYSED THE MEDIATOR ROLE PLAYED BY SOCIO-COGNITIVE THEORIES OF MOTIVATION (E.G., EXPECTANCY-VALUE AND ACHIEVEMENT GOAL THEORIES) IN THE RELATIONSHIP BETWEEN TEACHER-FAMILY SUPPORT AND STUDENT ENGAGEMENT.
Also, you state that “The present study considers this analysis from a new perspective by using the Educational Situation Quality Model (MOCSE, its acronym in Spanish) of Doménech-Betoret, [1-4] as a reference framework.” Could you be more specific on what you mean with the new perspective? How does the approach you take differ from the existing body of knowledge, how does this selected perspective add to that body? So I would, however, like to see more justifications for your research aim in the beginning, as this relates to the way you succeed in answering the ‘so what’ question regarding the research. I agree with the importance of understanding the link between support and engagement but would like to see the research gap presented in light of the existing body of knowledge.
MOCSE IS NOT SIMPLY A THEORETICAL MODEL THAT AIMS TO GUIDE RESEARCH IN THE EDUCATIONAL CONTEXT, LIKE MOST OF THE INSTRUCTIONAL MODELS FOUND IN THE LITERATURE, BUT ALSO PROVIDES A METHODOLOGICAL PROCEDURE FOR TEACHERS TO APPLY IT IN THE CLASSROOM. SO MORE THAN A CONCEPTUAL FRAMEWORK, IT IS ALSO A USEFUL TOOL THAT PROVIDES THE KEYS AND SYSTEMATIC GUIDELINES (EVALUATION, REFLECTION, INTERVENTION) SO THAT TEACHERS ARE ABLE TO DIAGNOSE THE MOTIVATIONAL DEFICIENCIES OF THEIR STUDENTS AND TO IDENTIFY THEIR POSSIBLE CAUSES IN TERMS OF DEMANDS AND SUPPORTS/RESOURCES. IT IS VALUABLE INFORMATION FOR DESIGNING IMPROVEMENT ACTIONS TO CORRECT DETECTED MOTIVATIONAL DEFICIENCIES AND TO IMPROVE ACADEMIC OUTCOMES. IMPROVEMENT ACTIONS CAN BE IMPLEMENTED EITHER DURING THE COURSE UNDERWAY OR THE NEXT COURSE (FOR MORE DETAILS, SEE DOMÉNECH-BETORET, 2018).
ANOTHER SPECIFIC ADVANTAGE OF THE MODEL FOR STUDYING SCHOOL ENGAGEMENT IS THAT MOCSE, BASED ON THE TEMPORAL CONCEPTION OF MOTIVATION (THEORY OF ACTION CONTROL, HECKHAUSEN AND KUHL, 1985; DÖRNYEI, 2000) DISTINGUISHES BETWEEN PREDECISIONAL MOTIVATION (DESIRE, INTENTION) AND POSTDECISIONAL MOTIVATION (EXECUTIVE OR VOLITIONAL MOTIVATION). THIS DISTINCTION ALLOWS PREVENTIVE MEASURES CENTERING ON STUDENTS’ ANTICIPATORY COGNITIVE MOTIVATORS TO BE ADOPTED (E.G., EXPECTATIONS) (FOR DETAILS, SEE DOMÉNECH-BETORET, 2018).
Heckhausen, H., and Kuhl, J. (1985). “From wishes to action: the dead ends and short cuts on the long way to action,” in Goal-directed Behaviour: The Concept of Action in Psychology, eds M. Frese and J. Sabini (Hillsdale, NJ: Lawrence Erlbaum), 134–160.
Dörnyei, Z. (2000). Motivation in action: Towards a process-oriented conceptualization of student motivation. British Journal of Educational Psychology, 70, 519-538.
Utilizing MOCSE framework. I believe I understand roughly what this framework is, but I would like to see the reasoning for adopting this.
BASED ON THE MOCSE FRAMEWORK, THE MAIN PURPOSE OF THIS STUDY IS TO BRIDGE THE JOB DEMANDS-RESOURCES MODEL (JD-R) AND TWO DOMINANT THEORIES OF ACHIEVEMENT MOTIVATION IN EDUCATION SUCH AS EXPECTANCY-VALUE AND ACHIEVEMENT GOAL THEORIES BY EXAMINING: A) HOW STUDENTS’ PERCEPTIONS OF SUPPORT/RESOURCES (from family and teacher) AND STUDENTS’ MOTIVATIONAL BELIEFS DERIVING FROM THE EXPECTANCY-VALUE THEORY RELATE TO STUDENT ENGAGEMENT; B) HOW STUDENTS’ PERCEPTIONS OF SUPPORT/RESOURCES (from family and teacher) AND STUDENTS’ MOTIVATIONA BELIEFS DERIVING FROM THE ACHIEVEMENT GOAL THEORY RELATE TO STUDENT ENGAGEMENT.
TWO THEORIES OF ACHIEVEMENT MOTIVATION
What is its added value when taking into consideration that you state your constructs are derived from other sources (Job Demands-Resources Model (JD-R) [5], the Expectancy-Value Theory [6], and the Achievement Goal Theory [7-10].)? How are you utilizing MOCSE in particular? My point for asking this relates to the presentation of Figure 1. My view is that the figure should present the framework of _your study_, and not the MOCSE framework since now there are a lot of concepts, relationships, and phases in the figure that is not in the focus of this study at hand, and also not being explained in the text at all. The figure should not have to explain itself, but all the central elements of this study should be opened in the text. A figure is a good way to present an overview of the framework of the study, but it should be then about this particular study at hand. Perhaps you could just refer to MOCSE and state that you present an adaptation of this model, and then focus on explaining how you do this. So I would suggest developing Section 1 of the paper in a way that would better address this particular study at hand and focus on its concepts.
I AGREE WITH REVIEWER 2'S SUGGESTION. SECTION 1 HAS BEEN REVISED AND AN ADPTATION OF THE MODEL FOR THIS SPECIFIC STUDY DEVISED.
FIGURE 1 PRESENTS AN OVERVIEW OF THE GENERAL FRAMEWORK OF THE MODEL CENTRED ON STUDENTS. THIS GENERAL FRAMEWORK WAS ADAPTED TO STUDY STUDENT ENGAGEMENT BASED ON THE MOCSE POSTUATES AND TAKING PREVIOUS PROPOSALS AS REFERENCES (DOMÉNECH-BETORET, ABELLÁN-ROSELLÓ, 2017; DOMÉNECH-BETORET, GÓMEZ-ARTIGA & ABELLÁN-ROSELLÓ, 2019). PLEASE SEE FIGURE 2.
AUTORES: Fernando Doménech Betoret, Amparo Gómez Artiga y Laura Abellán Roselló
TÍTULO: The Educational Situation Quality Model: A new tool to explain and improve academic achievement and course satisfaction.
REF. REVISTA: Front. Psychol., 18 July 2019 | https://doi.org/10.3389/fpsyg.2019.01692
AUTORES: FERNANDO DOMÉNECH BETORET Y LAURA ABELLÁN ROSELLÓ
TÍTULO: GUÍA PRÁCTICA PARA MEJORAR LA MOTIVACIÓN DEL ALUMNADO DE EDUCACIÓN SECUNDARIA Y FORMACIÓN PROFESIONAL.
REF. LIBRO: Col.lecció Educació. Publicaciones de la Universitat Jaume I, Castellón, 2017.
Figure 2. Adaptation of the MOCSE general framework to study student engagement.
Discussion
The problem regarding the ‘so what’ question I presented above now realizes in this section, as you state on p. 14 that “Although the motivation theme has been well-studied, with the present study we contribute a new approach using the MOCSE model [1-4] as a reference framework.” – This statement does not really tell anything about your contribution, it merely states that you have adopted a ‘new approach’. In what way is it novel (if you argue it is new, you need to show this in the light of the existing literature); and how does this new approach improve our understanding of student motivation/the research problem you have posed? I.e., ‘so what’?
FIRST, MOCSE, BASED ON THE TEMPORAL CONCEPTION OF MOTIVATION (THEORY OF ACTION CONTROL, HECKHAUSEN AND KUHL, 1985; DÖRNYEI, 2000) DISTINGUISHES BETWEEN PREDECISIONAL AND POSTDECISIONAL MOTIVATION, WHICH ENABLES A DEEPER STUDY OF MOTIVATION. THAT IS, IT ALLOWS US TO EXAMINE NOT ONLY THE ASPECT REFERRED TO THE DESIRE OR INTENTION THAT THE STUDENT HAS TO LEARN, BUT ALSO, IF THIS INTENTION IS IMPLEMENTED THROUGH CONCRETE ACTIONS (VOLITION OR EXECUTIVE MOTIVATION) DURING THE LEARNING PROCESS. EXECUTIVE MOTIVATION IS CLOSELY RELATED TO ENGAGEMENT (FILSECKER AND KERRES, 2014).
Dörnyei, Z. (2000). Motivation in action: Towards a process-oriented conceptualization of student motivation. British Journal of Educational Psychology, 70, 519-538.
Heckhausen, H., and Kuhl, J. (1985). “From wishes to action: the dead ends and short cuts on the long way to action,” in Goal-directed Behaviour: The Concept of Action in Psychology, Eds M. Frese and J. Sabini (Hillsdale, NJ: Lawrence Erlbaum), 134–160.
Filsecker, M. and Kerres, M. (2014). Engagement as a Volitional Construct: A Framework for Evidence-Based Research on Educational Games. Simulation & Gaming 45 (4-5), 450-470. https://doi.org/10.1177/1046878114553569
SECOND, GROUNDED IN THE SOCIO-COGNITIVE PERSPECTIVE OF MOTIVATION, EXPECTANCY-VALUE AND ACHIEVEMENT GOAL THEORIES ARE, ACCORDING TO THE LITERATURE, TWO OF THE MOST USED THEORIES IN THE STUDY OF ACHIEVEMENT MOTIVATION IN EDUCATION. HOWEVER, THE MOCSE MODEL INTRODUCES NEW CONSTRUCTS DERIVING FROM THE JOB DEMANDS-RESOURCES MODEL, TRADITIONALLY USED IN THE WORKPLACE AS AN ATTEMPT TO BETTER UNDERSTAND STUDENT MOTIVATION AND ENGAGEMENT. APPLYING THE JD-R MODEL TO THE SCHOOL CONTEXT FIRSTLY REQUIRES THOROUGHLY ANALYSING WHAT KIND OF LEARNING DEMANDS RELATED TO A SPECIFIC SUBJECT MATTER STUDENTS MUST ASSUME AND, SECONDLY, WHAT KIND OF SUPPORT (THE TEACHER, PEERS AND FAMILY, ETC.) STUDENTS SHOULD BE PROVIDED WITH DURING THE COURSE TO MEET THESE DEMANDS. HOW ALL THESE THREE THEORIES ARE RELATED TO STUDENT ENGAGEMENT IS THE MAIN GOAL OF THE PRESENT STUDY.
Methods:
On page 8 you state that “Those who did not complete all the questionnaires were eliminated, which considerably reduced the original sample size.” So how many respondents did you finally have?
THIS SENTENCE HAS BEEN CLARIFIED: “ ELIMINATING THOSE STUDENTS WHO DID NOT COMPLETE ALL THE QUESTIONNAIRES AT THE THREE STIPULATED TIMES LEFT THE SAMPLE WITH 267 PARTICIPANTS”
I hope you will find these observations helpful when developing your study further. Thank you once again for the opportunity to read your paper and wishing you all the best with your work!
APART FROM THE COMMENTS MADE HERE IN THE FORUM IN RESPONSE TO THE REVIEWERS’ SUGGESTIONS, I ALSO RESUBMIT THE MANUSCRIPT WITH THE CHANGES MADE HIGHLIGHTED IN GREEN TO FACILITATE THEIR PLACEMENT AND CHECKING.
Reviewer 3 Report
The article "Influence of teacher and family support on university student motivation and engagement", sent for review, raises an important issue of university education. Motivating actions taken by teachers and the family environment should be considered as one of the important factors in improving this teaching process. The presented research results make a significant contribution to the development of this field.
The contents of the "Introduction" indicate the great knowledge and scientific experience of the authors in the field of the described topic. The selection of research tools and statistical analysis is correct. The research results and their graphic presentation are clear and easy for the reader to perceive.
My only comment is chapter 1.5. [Objectives and hypotheses] which was limited to listing two hypotheses, ignoring the aim of the research (correctly formulated earlier in line 30). I would also suggest writing some research questions in this subsection.
Author Response
Comments and Suggestions for Authors
I VERY MUCH APPRECIATE THE EFFORT THAT REVIEWER 3’ HAS MADE TO REVIEW THIS ARTICLE.
The article "Influence of teacher and family support on university student motivation and engagement", sent for review, raises an important issue of university education. Motivating actions taken by teachers and the family environment should be considered as one of the important factors in improving this teaching process. The presented research results make a significant contribution to the development of this field.
The contents of the "Introduction" indicate the great knowledge and scientific experience of the authors in the field of the described topic. The selection of research tools and statistical analysis is correct. The research results and their graphic presentation are clear and easy for the reader to perceive.
My only comment is chapter 1.5. [Objectives and hypotheses] which was limited to listing two hypotheses, ignoring the aim of the research (correctly formulated earlier in line 30). I would also suggest writing some research questions in this subsection.
FOLLOWING REVIEWER 3’S SUGGESTION, I HAVE WRITTEN SOME RESEARCH QUESTIONS IN SUBSECTION 1.5. (OBJECTIVES AND HYPOTHESES) RELATED TO THE STUDY AIM: “HOW DO UNDERGRADUATE STUDENTS’ PERCEPTIONS OF THE SUPPORT THEY ARE PROVIDED WITH BY THE TEACHER INFLUENCE THEIR BELIEFS AND EXPECTANCIES?” “HOW DO UNDERGRADUATE STUDENTS’ PERCEPTIONS OF THE SUPPORT THEY ARE PROVIDED WITH BY FAMILY INFLUENCE THEIR BELIEFS AND EXPECTANCIES?” “HOW DO UNDERGRADUATE STUDENTS’ EXPECTANCIES EFFECT THEIR ENGAGEMENT?”
APART FROM THE COMMENTS MADE HERE IN THE FORUM IN RESPONSE TO THE REVIEWERS’ SUGGESTIONS, I ALSO RESUBMIT THE MANUSCRIPT WITH THE CHANGES MADE HIGHLIGHTED IN GREEN TO FACILITATE THEIR PLACEMENT AND CHECKING.
Round 2
Reviewer 1 Report
I accept all changes that have been made by the authors following my comments and suggestions.